# Instance Sequence Queries for Video Instance Segmentation with Transformers

**DOI:** 10.3390/s21134507

**Published:** 2021-06-30

**Authors:** Zhujun Xu, Damien Vivet

**Affiliations:** Institut Supérieur de l’Aéronautique et de l’Espace (ISAE-SUPAERO), University of Toulouse, 31400 Toulouse, France; damien.vivet@isae-supaero.fr

**Keywords:** video instance segmentation, transformer, query

## Abstract

Existing methods for video instance segmentation (VIS) mostly rely on two strategies: (1) building a sophisticated post-processing to associate frame level segmentation results and (2) modeling a video clip as a 3D spatial-temporal volume with a limit of resolution and length due to memory constraints. In this work, we propose a frame-to-frame method built upon transformers. We use a set of queries, called instance sequence queries (ISQs), to drive the transformer decoder and produce results at each frame. Each query represents one instance in a video clip. By extending the bipartite matching loss to two frames, our training procedure enables the decoder to adjust the ISQs during inference. The consistency of instances is preserved by the corresponding order between query slots and network outputs. As a result, there is no need for complex data association. On TITAN Xp GPU, our method achieves a competitive 34.4% mAP at 33.5 FPS with ResNet-50 and 35.5% mAP at 26.6 FPS with ResNet-101 on the Youtube-VIS dataset.

## 1. Introduction

Image instance segmentation has benefitted a lot from the development of Artificial Neural Networks over the last few years. As a fundamental element of many vision tasks, many methods [1,2,3,4,5,6,7,8,9] have been proposed with various structures and components, and achieve excellent performance with still images. However, many real world applications require the instance segmentation on video sequences. Not long ago, the video instance segmentation task was introduced by [10]. The goal of the VIS task is the simultaneous detection, segmentation and tracking of instances in videos [10]. The migration of still image methods to the VIS task is not trivial.

Previous works [3,10,11,12,13,14] mostly rely on two strategies. On one hand, some methods use complex post-processing to associate frame level instances. However, the track of instances can be easily disturbed by a few bad frame results. To recover the loss, sophisticated pipelines are established to aggregate supplementary features, which are time consuming. On the other hand, some methods pass all frames of a video together into the network, and let the network solve the instance association implicitly. However, these usually impose a limit of video resolution and length in order to fit the GPU memory. In addition, such offline methods are difficult to extend to online applications.

In this work, we propose instance sequence queries (ISQs) as an effective solution for VIS. We implement our method on a transformer based framework called DETR [15]. The transformer decoder is driven by the input queries. In DETR design, these input queries are learnt embeddings, which are fixed in inference time. By training the network with an extension of the bipartite loss to two frames, our method proves that the transformer decoder can actually adjust the queries, which we call ISQs, to track the same instance at inference. In our design, the same instance at different frames will be detected in the same query slot. The instance sequence is directly established without any extra data association. Figure 1 illustrates the architecture of our method. Furthermore, ISQs can be considered a flexible feature descriptor for corresponding instances. Thus, we can apply some simple mechanisms to produce robust results. For example, due to video qualities, such as motion blur or object occlusion, the latest appearance of the instance might not be the best representation for instance tracking. We adapt a mechanism for updating the ISQs with the best queries based on the confidence scores in the past. We also include a rollback mechanism for queries without an object. This mechanism serves as an NMS to erase duplicate detections without harming the consistency of instance sequences. Despite the efficacy of our concise framework, we believe our method can be integrated into many existing and future methods based on a transformer for a better performance.

## 2. Related Work

### 2.1. Video Instance Segmentation

Video instance segmentation is a task that involves classification, segmentation and tracking at the same time. Mask-Track RCNN [10] tackles the problem by extending the Mask RCNN [4] with a tracking branch to generate object features. A data association is established based on the similarity between features. SipMask [3] proposes an image instance segmentation architecture which preserves instance-specific spatial information. Similar to the Mask-Track RCNN [10], the model is extended with a tracking feature branch for the VIS task. Maskprop [11] uses predicted masks to crop image features, and to propagate the instance-specific features to improve the final results. IPDT [12] proposes a framework which uses instance-level and pixel-level embeddings in both tracking directions to calibrate segmentation results. STEm-Seg [13] models a video sequence as a 3D spatial-temporal volume. The network consists of a 3D CNN based encoder-decoder to process the backbone feature maps. VisTR [14] proposes an end-to-end structure based on transformers from the perspective of similarity learning. The network processes all image frames of a video clip in a single pass. Our method produces instance segmentation results frame-to-frame. We use the ISQs as a clue to preserve instance consistency; thus, no data association is required. Recently, Hwang et al. [16] proposed IFC as a competitive solution for the VIS task. Compared to other per-clip methods, IFC uses memory tokens to aggregate information from different frames, which reduces the computation and memory burden. Although the method can be applied to near-online inferences, it still needs soft-IOU post-processing to achieve instance matching. Since both IFC and our method are based on DETR, we believe our method can be integrated into IFC to yield a better matching result.

### 2.2. Transformer

The transformer was first introduced by [17]. The architecture based on attention mechanisms has rapidly become dominant in the Natural Language Processing (NLP) domain. DETR [15] develops a transformer encoder-decoder architecture for image object detection taska. With the conjunction of the bipartite matching loss and parallel decoding, it tackles object detection as a direct set prediction problems. Deformable DETR [18] combines DETR with the concept of the spatial sampling of deformable convolution. The attention module only attends to key sampling points around the reference, which leads to better performances and faster convergence of networks. Vision Transformer [19] builds a pure transformer architecture to solve the image classification problem without CNN. This further proves the performance of the transformer on vision tasks. Most of these works focus on optimising the inside of the transformer structure for vision applications, which leaves the transformer queries unexplored. TransTrack [20] constructs a query-key pipeline to solve the multiple-object tracking (MOT) task. Two parallel decoders are driven by learnt object queries and previous frame object feature queries to produce detection boxes and tracking boxes, respectively. An IoU matching is used to match two boxes’ predictions. TrackFormer [21] is quite close to our work, and uses transformer queries to process a video sequence in an auto-regressive fashion. They use the same learnt queries in each frame to generate new object detection. However, we discover the capacity of the transformer decoder to adjust queries for new objects by way of a different training procedure. Moreover, our method benefits from the flexibility of ISQs. By applying several simple mechanisms, our method provides a more robust performance. Both TransTrack and TrackFormer focus on the MOT17 and MOTS20 datasets, where the evaluation only accounts for pedestrians. We conduct our method on the Youtube-VIS dataset, which contains many more object classes.

## 3. Materials and Methods

Our method is built upon DETR [15]. In this section, we first introduce the architecture of the original DETR baseline. Then, we propose our network structure and inference pipeline to process the video sequences. Finally, we explain our extension of the bipartite matching loss to train our network.

### 3.1. Background: DETR

DETR [15] is an end-to-end image object detection framework built with transformers. The detection pipeline of DETR contains four main steps:A CNN backbone (e.g., ResNet [22]) takes an input image and extracts a compact feature representation;A transformer encoder encodes the image features with multi-head self-attention modules;A transformer decoder uses multi-headed attention mechanisms to decode the N learnt object queries;A feed forward network (FFN) computes the final prediction of the class and box for each object query.

The framework can be easily extended to solve the object segmentation task by adding a mask head. The mask head takes as input the output of the transformer decoder and generates attention maps for each predicted object. An FPN-style CNN follows, to generate the final mask of a high resolution.

DETR models object detection as a direct set prediction problem. The training procedure includes two steps: finding optimal bipartite matching between predicted and ground-truth objects, then calculating the specific loss between matched pairs. Both the class scores and the bounding boxes’ similarity are taken into account in the matching step. After finding the optimal bipartite matching, the network is trained with a Hungarian loss [23] composed of a negative log-likelihood for class prediction, a box loss and a mask loss.

We refer the reader to the original work on DETR [15] for further detail.

### 3.2. Instance Sequence Queries

#### 3.2.1. Definition

Given a video sequence of *T* frames, suppose there are *M* instances belonging to class set *C* in the video. For each instance, we denote its class with $c^{i}\in C$ and a sequence of binary segmentation masks with ${\left\{m_{t}^{i}\right\}}_{t=1}^{T}$. The $m_{t}^{i}$ can be an empty mask for the case where the instance *i* does not appear in frame *t*. Suppose an algorithm produces N predictions. Each prediction *j* contains a class ${\hat{c}}^{j}$, a confidence score ${\hat{s}}^{j}\in \left[0,1\right]$ and a sequence of predicted mask ${\left\{{\hat{m}}_{t}^{j}\right\}}_{t=1}^{T}$. The evaluation is based on mask IoU computation defined as:
(1)$$IoU\left(i,j\right)=\frac{\sum_{t=1}^{T}\left|m_{t}^{i}\cap {\hat{m}}_{t}^{j}\right|}{\sum_{t=1}^{T}\left|m_{t}^{i}\cup {\hat{m}}_{t}^{j}\right|}.$$

The DETR framework uses N learnt embeddings to drive the transformer decoder, which is called object queries. Each object query slot will produce one prediction. As presented in [15], each slot tends to learn a distribution specialized for spatial areas and object sizes. However, the position and size of the instance can change during time due to the instance movement or the camera motion. If we directly apply the object queries at each frame, the same instance might be detected in different slots. Thus, we introduce our design of ISQs, where the instance consistency over time is preserved by the query slot order. More specifically, the same instance at different frames will be detected by the query in the same slot.

#### 3.2.2. Inference Pipeline

Given a video clip, we initialize N embeddings similar to object queries, where N is larger than the number of instances in the video. For the first frame, we execute exactly the same pipeline as DETR. Then we pass the N outputs of the transformer decoder to the next frame as the ISQs. With the extension of bipartite matching loss to two frames, the decoder can learn to adjust the previous decoder outputs as a strong clue for the detection in the next frame. As shown in Figure 1, we follow this inference pipeline to produce segmentation results frame-to-frame. If a previous query detects no object, the decoder learns to use this query for upcoming new instances. This actually solves the case where an instance does not appear in the first frame of a video sequence. We will discuss this later in Section 5.

#### 3.2.3. Architecture

For the purposes of fair comparison, we mostly use the same architecture as DETR. We use ResNet [22] as the backbone. We use six layers of width 256 with eight attention heads for both encoder and decoder. The FFN is composed of a three-layer perceptron with an ReLU activation function and a linear projection layer. For the mask head, we also use a multi-head attention layer of width 256 with eight attention heads to generate the attention maps. The FPN-style CNN has exactly the same configuration as DETR. A difference is that we add a query attention module before passing the ISQs into decoder. The module is simply composed of a self-attention layer of width 256 with eight attention heads followed by a residual connection/dropout/layernorm. This additional module helps us to adapt the decoder outputs to the queries for the next frame.

#### 3.2.4. Instance Sequence Queries Update

Ideally, if an instance appears from one frame to the end of the video, our pipeline will update the corresponding query after each frame. However, if the instance quits the field of view before the last frame, its query slot starts to detect no object, and it might be used to detect new upcoming instances. Moreover, in the case of an occlusion or a false positive in one frame, its query might lead to a weak detection (or no object). If we update with this query, the re-detection of this instance can later be considered as a new instance that appears in another slot. To deal with such situations, we create a mechanism where each ISQ will be updated only when the new detection score is greater than or equal to the existing one with a tolerant margin of *p*. More specifically, let us denote by $Q={\left\{q_{i}\right\}}_{i=1}^{N}$ the existing ISQs after frame *t* and $\hat{P}={\left\{{\hat{p}}_{i}\right\}}_{i=1}^{N}$ the corresponding confidence score. We pass *Q* to process frame t+1 and receive $Q^{t+1}={\left\{q_{i}^{t+1}\right\}}_{i=1}^{N}$, ${\hat{P}}^{t+1}={\left\{{\hat{p}}_{i}^{t+1}\right\}}_{i=1}^{N}$ from the output. We update *Q* and *P* as:
(2)$$\left(q_{i},p_{i}\right)=\left\{\begin{array}{cc} \left(q_{i}^{t+1},q_{i}^{t+1}\right) & if\;{\hat{p}}_{i}^{t+1}\geq {\hat{p}}_{i}-\epsilon \\ \left(q_{i},p_{i}\right) & otherwise. \end{array}\right.$$

The tolerant margin ϵ ensures that the ISQs can still be updated despite a slight drop in score.

#### 3.2.5. Idle Queries Rollback

In most cases, the number of objects in one frame is less than the number of query slots. Some slots will be occupied by detected objects, while others stay idle with no object. As explained in [15], the bipartite matching implicitly imposes a suppression mechanism to remove duplicate predictions. If a video sequence has a long segment where there is almost no change, the occupied queries will keep suppressing the idle queries cumulatively. This will conduct a convergence of all idle queries as we observe. Since the decoder is permutation-invariant, it needs different input embedding to produce different results [15]. If a new object shows up later in the video sequence, all idle queries will make the same predictions which produces duplicate detections. In order to overcome this deficiency, thanks to the flexibility of ISQs, we introduce a rollback mechanism in the idle queries. In detail, if a query switches from idle to occupied with a significant increase in prediction score greater than 0.3, we conduct an NMS with a threshold of 0.9. If the prediction of this query is suppressed by NMS, we do a rollback to this query which keeps it in idle status. This helps to stop the duplicate detections from keeping active for the rest of the video sequence.

### 3.3. Training

We trained our model with the Youtube VIS dataset. For each training iteration, we used two images randomly sampled from the same video sequence. We denote by $I^{t}$ and $I^{t+\tau }$ the two sampled images. The ground-truth sets are denoted as ${\left\{y_{i}^{t}\right\}}_{i=1}^{N}$ and ${\left\{y_{i}^{t+\tau }\right\}}_{i=1}^{N}$, respectively. Both sets are padded with ⌀ as no object. To be clear, the $y_{i}^{t}$ and $y_{i}^{t+\tau }$ represent the same instance *i* at frame *t* and frame t+τ. We pass the $I^{t}$ to the network with the learnt embeddings as queries. This step is exactly the same as the DETR single image routine and we obtain the prediction set ${\left\{{\hat{y}}_{i}^{t}\right\}}_{i=1}^{N}$. We extract the outputs from the transformer decoder and use them as the ISQs for the process of frame t+τ. Thus, we obtain the prediction set ${\left\{{\hat{y}}_{i}^{t+\tau }\right\}}_{i=1}^{N}$. Similarly, the ${\hat{y}}_{i}^{t}$ and ${\hat{y}}_{i}^{t+\tau }$ represent predictions in the same query slot *i* at frame *t* and frame t+τ.

In order to adapt our training procedure, we extended the bipartite matching loss to two frames. Given the definitions above, we tried to find a permutation of N elements $\tilde{\sigma }$ to minimize the matching loss:
(3)$$\tilde{\sigma }=\underset{\sigma }{arg min}\sum_{i}^{N}L_{match}\left(y_{i}^{t},{\hat{y}}_{\tilde{\sigma }\left(i\right)}^{t},y_{i}^{t+\tau },{\hat{y}}_{\tilde{\sigma }\left(i\right)}^{t+\tau }\right).$$

The ground-truth instance $y_{i}^{t}$, $y_{i}^{t+\tau }$ contains classes $c_{i}^{t}$, $c_{i}^{t+\tau }$ and bounding boxes $b_{i}^{t}$, $b_{i}^{t+\tau }$. We emphasize that the situation exists wherein an instance only appears in one of the two frames. In that case, only one of $c_{i}^{t}$, $c_{i}^{t+\tau }$ is no object. Actually, it is crucial to include such a situation in our training plan. The influence is presented in Section 5.2. For the prediction with index $\tilde{\sigma }\left(i\right)$, the scores of the class are denoted as ${\hat{p}}_{\tilde{\sigma }\left(i\right)}^{t}\left(c_{i}^{t}\right)$ and ${\hat{p}}_{\tilde{\sigma }\left(i\right)}^{t+\tau }\left(c_{i}^{t+\tau }\right)$. The predicted boxes are denoted as ${\hat{b}}_{\tilde{\sigma }\left(i\right)}^{t}$ and ${\hat{b}}_{\tilde{\sigma }\left(i\right)}^{t+\tau }$. The matching loss is defined as:
(4)$$\begin{array}{c} L_{match}\left(y_{i}^{t},{\hat{y}}_{\tilde{\sigma }\left(i\right)}^{t},y_{i}^{t+\tau },{\hat{y}}_{\tilde{\sigma }\left(i\right)}^{t+\tau }\right)=-{𝟙}_{\left\{c_{i}^{t}\neq ⌀\;or\;c_{i}^{t+\tau }\neq ⌀\right\}}\left[{\hat{p}}_{\tilde{\sigma }\left(i\right)}^{t}\left(c_{i}^{t}\right)+{\hat{p}}_{\tilde{\sigma }\left(i\right)}^{t+\tau }\left(c_{i}^{t+\tau }\right)\right]\\ +{𝟙}_{\left\{c_{i}^{t}\neq ⌀\right\}}L_{box}\left(b_{i}^{t},{\hat{b}}_{\tilde{\sigma }\left(i\right)}^{t}\right)+{𝟙}_{\left\{c_{i}^{t+\tau }\neq ⌀\right\}}L_{box}\left(b_{i}^{t+\tau },{\hat{b}}_{\tilde{\sigma }\left(i\right)}^{t+\tau }\right). \end{array}$$

The ${𝟙}_{\left\{c_{i}\neq ⌀\right\}}$ is an indicator function that equals 1 if $c_{i}\neq ⌀$ and equals 0 otherwise. The $L_{box}\left(b_{i},{\tilde{b}}_{\tilde{\sigma }\left(i\right)}\right)$ is composed of a $\ell_{1}$ loss and an IoU loss [24] to adapt for objects of different scales. With two hyperparameters, the box loss is defined as:(5)$$L_{box}\left(b_{i},{\hat{b}}_{\tilde{\sigma }\left(i\right)}\right)=\lambda_{L1}{∥b_{i}-{\hat{b}}_{\tilde{\sigma }\left(i\right)}∥}_{1}+\lambda_{iou}L_{iou}\left(b_{i},{\tilde{b}}_{\tilde{\sigma }\left(i\right)}\right).$$

The training loss is the sum of the Hungarian loss applied to the two pairs of ground-truth and prediction . It is defined as below:(6)$$\begin{array}{cc} \; & L_{Hung}\left(y^{t},{\hat{y}}^{t},y^{t+\tau },{\hat{y}}^{t+\tau }\right)=\\ & \sum_{i=1}^{N}\left[-\log{\hat{p}}_{\tilde{\sigma }\left(i\right)}^{t}\left(c_{i}^{t}\right)+{𝟙}_{\left\{c_{i}^{t}\neq ⌀\right\}}L_{box}\left(b_{i}^{t},{\hat{b}}_{\tilde{\sigma }\left(i\right)}^{t}\right)\right.+{𝟙}_{\left\{c_{i}^{t}\neq ⌀\right\}}L_{mask}\left(m_{i}^{t},{\hat{m}}_{\tilde{\sigma }\left(i\right)}^{t}\right)\\ & -\log{\hat{p}}_{\tilde{\sigma }\left(i\right)}^{t+\tau }\left(c_{i}^{t+\tau }\right)+{𝟙}_{\left\{c_{i}^{t+\tau }\neq ⌀\right\}}L_{box}\left(b_{i}^{t+\tau },{\hat{b}}_{\tilde{\sigma }\left(i\right)}^{t+\tau }\right)+{𝟙}_{\left\{c_{i}^{t+\tau }\neq ⌀\right\}}L_{mask}\left(m_{i}^{t+\tau },{\hat{m}}_{\tilde{\sigma }\left(i\right)}^{t+\tau }\right)], \end{array}$$
where the $m_{i}^{t}$ and ${\tilde{m}}_{\tilde{\sigma }\left(i\right)}^{t}$ are the ground-truth binary segmentation mask and the predicted mask at frame *t*. With a hyperparameter, the mask loss is defined as a combination of the DICE/F-1 loss [25] and Focal loss [26]:(7)$$L_{mask}\left(m_{i},{\tilde{m}}_{\tilde{\sigma }\left(i\right)}\right)=\lambda_{mask}\left[L_{Dice}\left(m_{i},{\tilde{m}}_{\tilde{\sigma }\left(i\right)}\right)\right.\left.+L_{focal}\left(m_{i},{\tilde{m}}_{\tilde{\sigma }\left(i\right)}\right)\right].$$

## 4. Results

### 4.1. Dataset

In order to compare with other methods, we used the Youtube-VIS 2019 version [10] dataset to evaluate our method. The dataset has 2238 training videos, 302 validation videos and 343 test videos. The dataset has a category label set including 40 common objects such as people, animals and vehicles. Each video sequence contains at most 36 frames. Each frame is a temporal downsampling of the original sequence with an interval of five frames, which leads to a greater variation between consecutive frames. This can actually simulate the scenarios with high speed motion. Such scenarios usually require the algorithm to be more robust. As the test set evaluation server is closed, we followed most previous works [3,10,11,13,14] and evaluated our method on the validation set. The evaluation metrics are Average Precision (AP) calculated based on multiple intersection-over-union (IoU) thresholds and Average Recall (AR) defined as the maximum recall given some fixed number of segmented instances per video.

### 4.2. Implementation Details

We used ResNet-50 as our backbone network, unless otherwise noted. We used the same hyperparameters as DETR to build the losses. We set the number of ISQs *N* to 10, which is enough for the dataset. The model was trained with the AdamW optimizer [27]. We set the initial transformer’s learning rate to $10^{-4}$, the backbone’s to $10^{-5}$ and the weight decay to $10^{-4}$. The backbone network and transformers were initialized with DETR pre-trained weights on MS COCO [28]. We trained the networks for 50 epochs and the learning rate drops by a factor of 10 at the 30th epoch. The model was trained on two Titan Xp GPUs of 12G RAM for about 20 h, with a batch size of 14. All frames were resized to 640×360. Random horizontal flip was used as data augmentation.

### 4.3. Main Results

We show the results of our method and the state-of-the-art in Table 1. Our method produces a comparative performance in terms of both accuracy and speed. We gain about four points in AP against the MaskTrack R-CNN [10] with faster FPS. We also slightly outperform SipMask [3] in both AP and runtime. The gap of accuracy between our method and MaskProp [11] is mainly due to the fact that Maskprop combines multiple networks and post-processing strategies which are actually time-consuming. For STEm-Seg [13] and VisTR [14], both methods process the entire video sequence at the same time. Such methods usually impose limits on video resolution and length, which are difficult to extend to online applications. Nonetheless, our method achieved better than, or close to the same, performances without such limits. The IFC [16] shows great speed–accuracy balance under near-online settings. Given the fact that IFC and our method share much in common in terms of structure, we believe both methods can be merged. By replacing the object queries with our ISQs design and extending the training procedure to 2T frames, a better result can be expected.

## 5. Discussion

In this section, we conduct ablation studies to explain the necessity of different components in our method.

### 5.1. DETR Baseline

We compare our methods with the DETR baseline. On the one hand, we try to adapt DETR for VIS tasks with post-processing. As mentioned above, the Youtube VIS dataset is a temporal down sampling version of the original videos. This largely increases the instance displacement between consecutive frames. Simply applying an IoU association will actually break an instance track into several pieces. Each piece of track will be considered as a different instance in the evaluation. Such duplicates of tracks hurts the AP of VIS. On the other hand, we directly use the query slot index of DETR as an instance index to generate VIS results. To our surprise, this can actually conduct 32 points in AP. However, the object queries of DETR show spatial and object size distribution. As shown in Figure 2, once the instance passes the boundary of a query slot, it leads to a switch ID error. However, our methods adapt the queries to the instance, which keeps the detection of instances in the same query slot.

### 5.2. New Instance Detection

An instance may not appear from the beginning of a video sequence. The ability to detect new instances is an essential requirement for VIS algorithms. In our training procedure, we include situations where some instances only appear in one of the two training frames. This is actually the key to new instance detection. We use two images randomly sampled from the video for training. This actually increases the occurrence probability of the above situations as training samples. We conduct an ablation study by excluding these situations. As a result, we observe a significant performance drop of about 6 points in AP. The network can hardly achieve the new instance detection much longer.

### 5.3. Ability to Adjust Queries

Object queries are very sensitive in the DETR framework. The arbitrary modification to learnt queries leads to great damage to performance. However, in our study, we find that the transformer decoder actually has the ability to adjust ISQs. We conduct an exaggerated experiment to prove it. We analyze this ability by randomly initializing the ISQs in the first frame. As expected, the random initial queries lead to a worse performance. However, we notice that the randomization only harms the results of several frames at the beginning. The transformer decoder can correct these random queries progressively. We exclude the results of the first several frames of each video sequence and compare the AP performance between the model with random initial queries and learnt queries. As presented in Table 2, the difference decreases as we exclude more frames.

### 5.4. Query Update

We presented our query update mechanism in Section 3.2.4. This mechanism is to recover bad single frame results. We show a typical example in Figure 3. At frame 13, due to insufficient feature extraction, the network fails to detect the parrot on the right side. Query 8 (in red) tries to capture the parrot on the left side asa prediction with a low score, which is a false positive (FP). At frame 14, if we update query 8 with output from frame 13, this query will contain FP information. Thus, we cannot detect the correct instance. In Figure 4, we plot the prediction score of all queries at each frame. We notice that the score of query 8 (red line) keeps decreasing and finally becomes an idle query which detects the background. At frame 18, the network tries to recapture this instance as a new detection. Finally, query 4 suppresses query 8. Thus, the instance is tracked in query 4 for the rest of the video. However, if we apply the conditional update mechanism, we can avoid the above results. At frame 14, since the detection score has a large drop, the query will update with output from frame 12. Then we recapture the instance in the same query slot. The instance keeps tracking in query 8 without new object detection being triggered. We present performance results with the different margin ϵ in Table 3. This mechanism actually adds an extra 0.8 AP to our model.

This mechanism also reveals an interesting phenomenon. As we know, in the attention module, the queries will multiply with keys and pass through a softmax function to generate a scaled weight for the values. Intuitively, to modify only parts of queries will affect the others’ results. However, the entire frame is robust to handling such modifications, even if we do not deliberately train this ability.

### 5.5. Idle Queries Rollback

We introduced the idle queries rollback mechanism in Section 3.2.5. This mechanism serves to avoid duplicate detections caused by the cumulative suppression of idle queries. We show an example in Figure 5. From frame 1 to frame 9, there is almost no change in the video. At frame 10, a person starts to enter into the field of view. Multiple queries become active to this new object. These duplicate detections will keep existing until the end of the video. However, when applying the rollback mechanism, all duplicated queries come back to idle status. The duplicates are removed at Frame 11. We emphasize that our mechanism is different from simply applying NMS. As all duplicate predictions are close in score, we cannot ensure that the survivor after NMS is the same query in the video sequence. Our method cuts the source of the problem early. Although this situation is rare in the YouTube-VIS validation set, it does not really hurt the AP results. However, in some online applications where the video sequence can be much longer, the effect of the mechanism will be more obvious.

## 6. Conclusions

In this paper, we introduce a solution for the Video Instance Segmentation task. Compared with state-of-the-art methods, our framework is based on a transformer and achieves a competitive performance. Our simple and clean frame-to-frame pipeline establishes an instance sequence without any complex data association. We also exploit the extensibility and flexibility of transformer queries. We believe more mechanisms and structures can be applied to transformer queries to extend the functionality and enhance the performance. We hope our work can provide inspiration for other approaches that apply a transformer in computer vision.

## Figures and Tables

**Figure 1 sensors-21-04507-f001:**
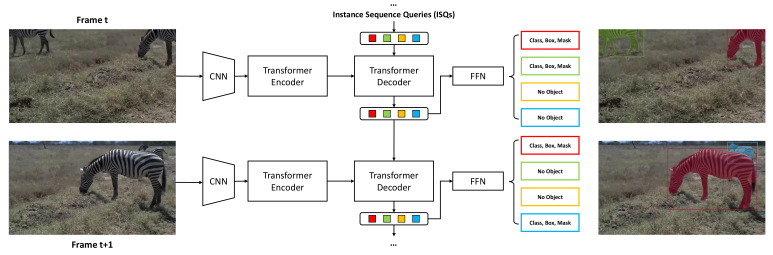
Architecture of our method. The transformer decoder uses ISQs as a clue for capturing similar instances at frame t. Meanwhile, the decoder outputs are adjusted based on the instances’ best appearances and become the input queries for the next frame.

**Figure 2 sensors-21-04507-f002:**
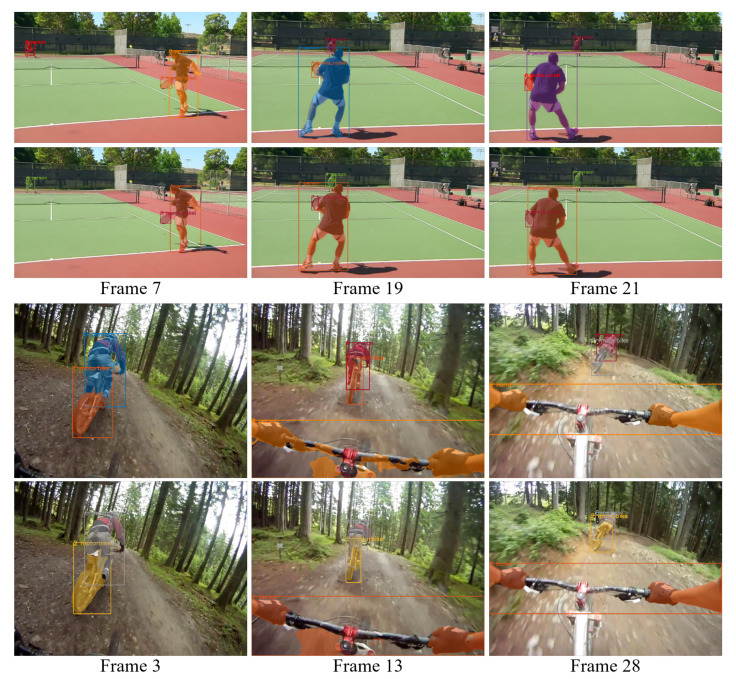
Result comparison between our method and the baseline DETR. At each frame, the upper result is conducted by DETR, the lower is our solution. The same color represents predictions from the same query slot. In the first line, as the person and the tennis racket move from the right side to the left in camera. DETR detects them with different query slots. In the second line, the size of the person changes between different frames. DETR uses different queries to detect him. However, in our framework, despite the spatial and size changes, the same instance is detected in the same query slot.

**Figure 3 sensors-21-04507-f003:**
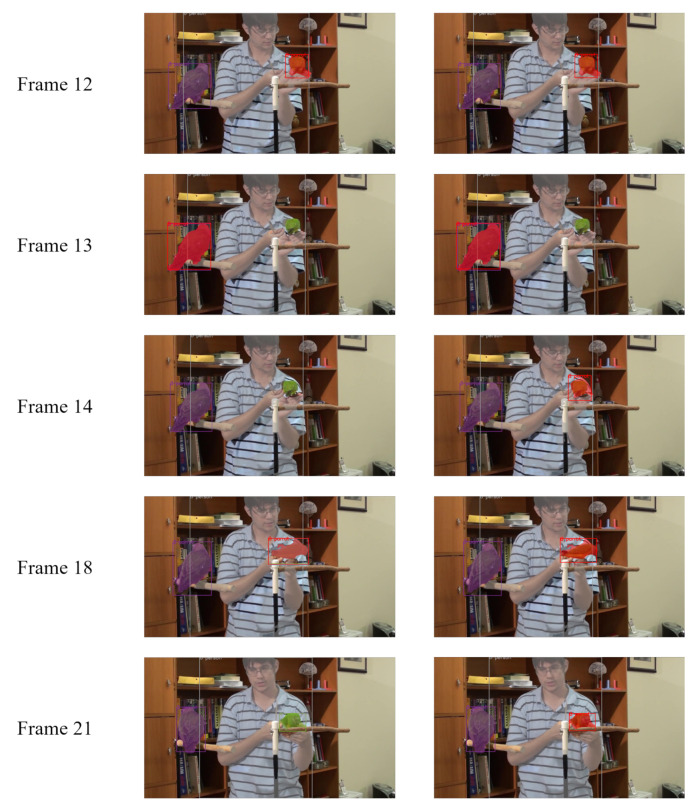
An example of the query update mechanism’s effect. At each frame, the left image is the result as we update ISQs after each frame. The right image is the result when we apply our update mechanism. The same color represents predictions from the same query slot. We illustrate detections with scores greater than 0.3.

**Figure 4 sensors-21-04507-f004:**
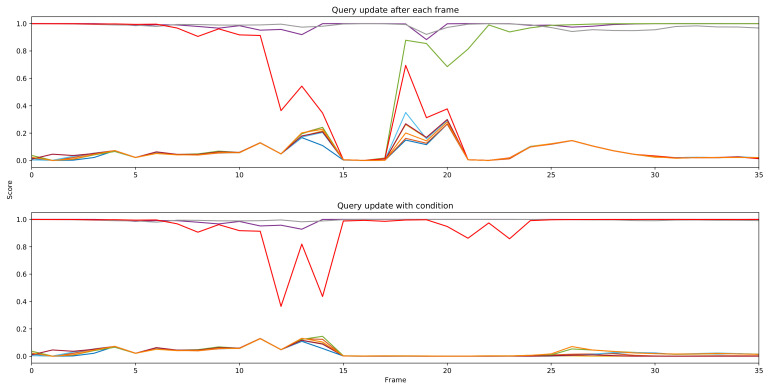
Prediction scores of all 10 queries at each frame of a video sequence. Different queries are represented by different colors.

**Figure 5 sensors-21-04507-f005:**
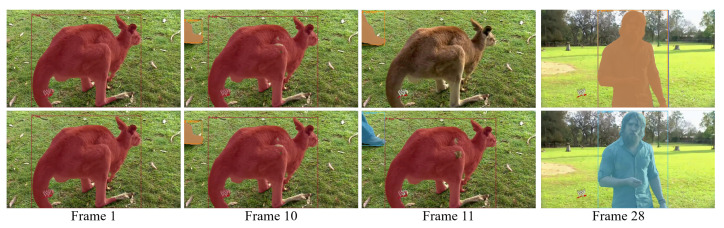
An example of the idle query rollback mechanism. At each frame, the upper/lower image is the result without/with this mechanism.

**Table 1 sensors-21-04507-t001:** Performance comparison on the Youtube-VIS validation set. All models are pre-trained on MS COCO. We test MaskTrack RCNN, SipMask and our method on the same machine. For other methods, we report the results from the original papers.

Method	Backbone	Resolution	AP	${\mathrm{AP}}_{50}$	${\mathrm{AP}}_{75}$	${\mathrm{AR}}_{1}$	${\mathrm{AR}}_{10}$	FPS
MT RCNN [10]	ResNet-50	640 × 360	30.3	51.1	32.6	31.0	35.5	20.0
SipMask [3]	ResNet-50	640 × 360	33.7	54.1	35.8	35.4	40.1	30.0
MaskProp [11]	ResNet-50	640 × 360	40.0	−	42.9	−	−	−
MaskProp [11]	ResNet-101	640 × 360	42.5	−	45.6	−	−	−
STEm-Seg [13]	ResNet-50	640∼1196	30.6	50.7	33.5	31.6	37.1	−
STEm-Seg [13]	ResNet-101	640∼1196	34.6	55.8	37.9	34.4	41.6	−
VisTR [14]	ResNet-50	540 × 300	34.4	55.7	36.5	33.5	38.9	30.0
VisTR [14]	ResNet-101	540 × 300	35.3	57.0	36.2	34.3	40.4	27.7
IFC (online) [16]	ResNet-50	640 × 360	41.0	62.1	45.4	43.5	52.7	46.5
IFC (offline) [16]	ResNet-50	640 × 360	42.8	65.8	46.8	43.8	51.2	107.1
IFC (offline) [16]	ResNet-101	640 × 360	44.6	69.2	49.5	44.0	52.1	89.4
Our Method	ResNet-50	640 × 360	34.4	54.9	37.7	33.3	38.1	33.5
Our Method	ResNet-101	640 × 360	35.5	56.6	39.2	34.8	39.9	26.6

**Table 2 sensors-21-04507-t002:** Comparison of results with random initial queries and learnt initial queries. We notice that the difference between two initializations vanishes as we exclude more frames at the beginning. Since the valid annotation is not publicly available, we can only evaluate the performance on the dataset official server. That is the main reason for the AP decrease.

Excluded Frames	Random Initial	Learnt Initial
Non exclusion	32.7	34.1
1	29.0	30.5
1 to 3	22.1	23.2
1 to 5	16.9	17.3
1 to 7	12.1	12.4
1 to 9	7.3	7.5

**Table 3 sensors-21-04507-t003:** Ablation study of our query update mechanism; w/o refers to results without this update mechanism.

Margin ϵ	w/o	0	0.05	0.10	0.15	0.20
AP	33.6	33.1	34.4	34.1	34.2	34.2

## Data Availability

The Youtube-VIS dataset [10] is publicly available.

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
