# Peer review of "Instance Sequence Queries for Video Instance Segmentation with Transformers"

_sensors, 2021, doi:10.3390/s21134507_

Round 1
Reviewer 1 Report
I have the following suggestions for the presented article:
1. In the abstract, mention what hardware or environment was used to obtain the 33.5 FPS and 26.6 FPS processing speed.
2. Please emphasize which dataset was used for training at the beginning of the “3.3. Training” section to be more explicit for the reader.
3. In the “4.1. Dataset” section, specify the exact version of the Youtube-VIS dataset used for training (https://youtube-vos.org/dataset/vis/).
4. As the “4.2. Implementation details” section mentions, the training was performed on a local machine with Titan Xp GPUs. Therefore, I recommend running the trained model on a local machine on the test video set instead of the validation set, in which case the results are prone to overfitting.
5. From the research repeatability perspective, mentioning the training time of the model is a good practice.
6. In the “4.3. Main results” section, please specify the hardware or environment configuration on which the networks were tested. The practicality of the presented novel architecture mostly depends on its deployability.
Author Response
Thank you for your resourceful suggestions:
1. We will add the environment in Abstract.
2. We will specify the dataset in section 3.3
3. We will specify that we used the version 2019 of YoutubeVIS in section 4.1
4. We use the 2019 to evaluate our performance in order to compare with other methods. However, the valid and test annotations are not provided. The evaluation is executed on official server (by uploading results files). Moreover, the test server is closed, thus we can only evaluate on valid set. We adapt the same way of evalutation as other paper, thus we think it is a fair comparision with other methods.
5. We will add the 20 hours training time in the article
6. For MaskTrackRCNN and SipMask, we test on the same configuration as our method. For MaskProp, STEm-Seg and VisTR, due to their complicity or no code is provided, we report the results from the original papers. We will specifiy these in our article.
Reviewer 2 Report
This paper proposes a new framework of Instance Sequence Queries (ISQs) for the problem of Video Instance Segmentation, which is built upon transformers (DETR). Several mechanisms such as the design of a new training procedure, ISQs update, and idle query rollback are introduced to improve the robustness of the proposed framework. ISQs and other state-of-the-art approaches for Video Instance Segmentation are evaluated with the Youtube-VIS dataset. Additional ablation studies are also presented to demonstrate the improvements from the mechanisms introduced in ISQs.
This paper includes a comprehensive comparison between the proposed ISQs framework and other popular approaches for Video Instance Segmentation, demonstrating competitive performance in the Youtube-VIS dataset. In addition, this paper presents an extensive analysis for the ablation studies, which clearly explains the benefits from the introduced techniques (e.g., ISQs update, idea query rollback).
The main area of improvements for the current presentation is to better clarify the contributions and the significance of the proposed framework:
- The idea of applying transformers to Video Instance Segmentation is not new. The authors also mentioned some previous works based on transformers (e.g., [14], [19-20]). From the current presentation, it is not very clear what the advantages are from the proposed framework compared to the previous works. Is the main contribution of the paper coming from the mechanisms introduced in ISQs (e.g., ISQs update, idea query rollback)? However, these mechanisms are not discussed in the Introduction section. More explanations and discussions are needed in the Introduction and Related Work sections to better clarify the contributions.
- In addition, there is some recent work for Video Instance Segmentation [1*]. [1*] is also based on DETR and could achieve promising near-online performance by flexibly adjusting the number of frames in the video clip. How does the proposed ISQs framework compare with the approach in [1*]? What are the strengths of the proposed framework? The authors should consider including [1*] in the discussion and experiments for the comparison purpose.
[1*] Hwang, Sukjun, et al. "Video Instance Segmentation using Inter-Frame Communication Transformers." arXiv preprint arXiv:2106.03299 (2021)
Moreover, the Introduction section can be further improved. The introduction should provide a more clear description of the proposed method. For example, ISQs should probably be defined/explained more clearly in the introduction section. In addition, as mentioned above, several mechanisms (e.g., ISQs update, idle query rollback) seem quite important for robustness but none of these mechanisms are mentioned in the Introduction section. To provide a more complete picture of the proposed framework, it is better to briefly discuss these mechanisms in the Introduction section.
Lastly, the organization of the paper could be also improved. For instance, the description of some mechanisms (e.g., idle query rollback) comes very late in the current presentation. The detailed description/discussions are presented in the Discussion section (Section 5). What’s more, the Discussion section (section 5) mixes the description of the mechanisms introduced in ISQs and the experiment results/analysis. The authors could consider separating these two parts (e.g., move the description of the mechanisms to earlier sections, i.e., Section 3) for better organization.
Other detailed comments/notes:
- There are a few places in this paper where some citations appear after a period, e.g., page 1 line 20, page 2 line 89, page 10 line 260. The authors should consider fixing them.
- It seems that the evaluation is done in Youtube-VIS 2019. What about the results with the more recent Youtube-VIS 2021 dataset?
Author Response
Thank you for your resourceful suggestions.
The main contribution of our work is 1) the new training procedure to enable the ISQ for VIS task. 2) the mechanisms like ISQ update, rollback. We will add some explainations in introduction section.
Thank you for your information about the recent work for VIS with DETR baseline which is quite interesting. We missed it as it is really a lastest work. We will add comparasion with this work.
We will also add more explaination in introduction to quickly give the readers a brief understanding of our structure.
We will rearrange section 5 for better explaination.
Thank you for your kind reminder about the period, we will fix them.
We evaluate our work in Youtube-VIS 2019 for a fair comparasion with other works. We have not noticed the latest version of the dataset. We try to do training and evalutation right now. But we are not sure whether we can catch up the deadline. We will try our best to do that.